# LEARNING TO REPEAT:
# FINE GRAINED ACTION REPETITION FOR
# DEEP REINFORCEMENT LEARNING

**Sahil Sharma, Aravind S. Lakshminarayanan, Balaraman Ravindran**
Indian Institute of Technology, Madras
Chennai, 600036, India
{sahil, ravi}@cse.iitm.ac.in
aravindsrinivas@gmail.com

## ABSTRACT

Reinforcement Learning algorithms can learn complex behavioral patterns for sequential decision making tasks wherein an agent interacts with an environment and acquires feedback in the form of rewards sampled from it. Traditionally, such algorithms make decisions, i.e., select actions to execute, at every single time step of the agent-environment interactions. In this paper, we propose a novel framework, Fine Grained Action Repetition (FiGAR), which enables the agent to decide the action as well as the time scale of repeating it. FiGAR can be used for improving any Deep Reinforcement Learning algorithm which maintains an explicit policy estimate by enabling temporal abstractions in the action space. We empirically demonstrate the efficacy of our framework by showing performance improvements on top of three policy search algorithms in different domains: Asynchronous Advantage Actor Critic in the Atari 2600 domain, Trust Region Policy Optimization in Mujoco domain and Deep Deterministic Policy Gradients in the TORCS car racing domain.

## 1 INTRODUCTION

Reinforcement learning (RL) is used to solve goal-directed sequential decision making problems wherein explicit supervision in the form of correct decisions is not provided to the agent, but only evaluative feedback in the form of the rewards sampled from the environment. RL algorithms model goal-directed sequential decision making problems as Markov Decision Processes (MDP) [Sutton & Barto (1998)]. However, for problems with an exponential or continuous state space, tabular RL algorithms that maintain value or policy estimates for every state become infeasible. Therefore, there is a need to be able to generalize decision making to unseen states. Recent advances in representation learning through deep neural networks provide an efficient mechanism for such generalization [LeCun et al. (2015)]. Such a combination of representation learning through deep neural networks with reinforcement learning objectives has shown promising results in many sequential decision making domains such as the Atari 2600 domain [Bellemare et al. (2013); Mnih et al. (2015); Schaul et al. (2015); Mnih et al. (2016)], Mujoco simulated physics tasks domain [Todorov et al. (2012); Lillicrap et al. (2015)], the Robosoccer domain [Hausknecht et al. (2016)] and the TORCS domain [Wymann et al. (2000); Mnih et al. (2016)]. Often, MDP settings consist of an agent interacting with the environment at discrete time steps. A common feature shared by all the Deep Reinforcement Learning (DRL) algorithms above is that they repeatedly execute a chosen action for a fixed number of time steps $k$. If $a_t$ represents the action taken at time step $t$, then for the said algorithms, $a_1 = a_2 = \cdots = a_k$, $a_{k+1} = a_{k+2} = \cdots = a_{2k}$ and in general $a_{ik+1} = a_{ik+2} = \cdots = a_{(i+1)k}$, $i \geq 0$. Action repetition allows these algorithms to compute the action once every $k$ time steps and hence operate at higher speeds, thus achieving real-time performance. This also offers other advantages such as smooth action policies. More importantly, as shown in Lakshminarayanan et al. (2017) and Durugkar et al. (2016), macro-actions constituting the same action repeated $k$ times could be interpreted as introducing temporal abstractions in the induced policies thereby enabling transitions between temporally distant advantageous states.

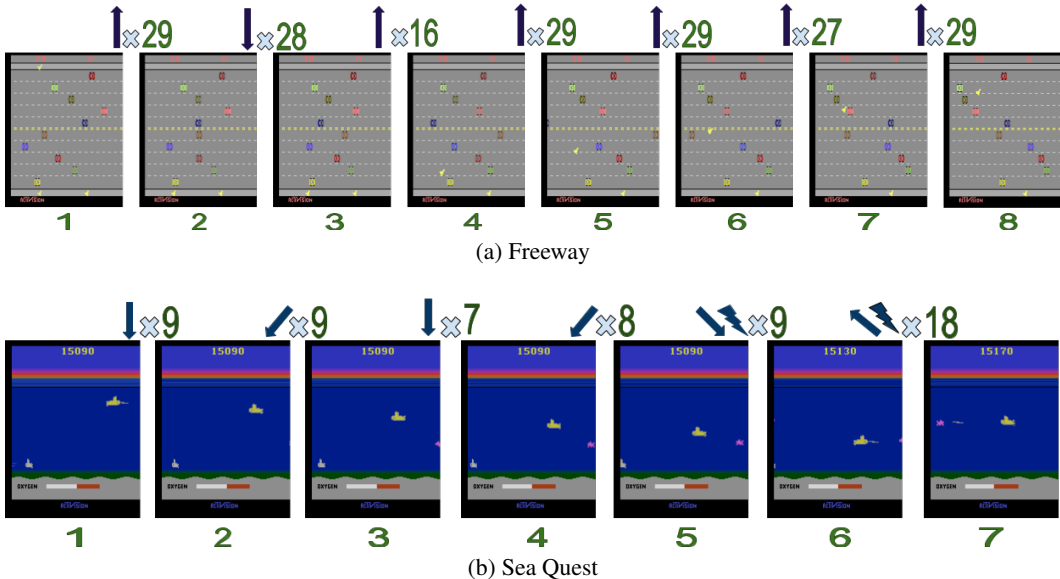

Figure 1: FiGAR induces temporal abstractions in learnt policies. The arrows indicate the action executed between the frames and the numbers depict the number of time steps for which the action was repeated. The thunder bolt corresponds to the firing action. An arrow alongside a thunderbolt corresponds to the action (arrow+fire). In the figure (a), the agent learns to execute down operation (which is equivalent to a no-op in this particular state, in this game) until a traveling car passes by and then executes temporally elongated actions to complete the task, skillfully avoiding the red car in the $7^{th}$ frame. In figure (b) the agent catches a glimpse of a pink opponent towards bottom right in the $2^{nd}$ frame and executes temporally elongated actions to intercept and kill it (in the $6^{th}$ frame).

The time scale for action repetition has largely been static in DRL algorithms until now [Mnih et al. (2015; 2016); Schaul et al. (2015)]. Lakshminarayanan et al. (2017) are the first to explore dynamic time scales for action repetition in the DRL setting and show that it leads to significant improvement in performance on a few Atari 2600 games. However, they choose only two time scales and the experiments are limited to a few representative games. Moreover the method is limited to tasks with a discrete action space.

We propose FiGAR, a framework that enables any DRL algorithm regardless of whether its action space is continuous or discrete, to learn temporal abstractions in the form of temporally extended macro-actions. FiGAR uses a structured and factored representation of the policy whereby the policy for choosing the action is decoupled from that for the action repetition selection. Note that deciding actions and the action repetitions independently enables us to find temporal abstractions without blowing up the action space, unlike Vezhnevets et al. (2016) and Lakshminarayanan et al. (2017). The contribution of this work is twofold. First, we propose a generic extension to DRL algorithms by coming up with a factored policy representation for temporal abstractions (see figure 1 for sequences of macro actions learnt in 2 Atari 2600 games). Second, we empirically demonstrate FiGAR's efficiency in improving policy gradient DRL algorithms with improvements in performance over several domains: 31 Atari 2600 games with Asynchronous Advantage Actor Critic [Mnih et al. (2016)], 5 tasks in MuJoCo Simulated physics tasks domain with Trust Region Policy Optimization [Schulman et al. (2015)] and the TORCS domain with Deep Deterministic Policy Gradients [Lillicrap et al. (2015)].

## 2  RELATED WORK

Our framework is centered on a very general idea of *only deciding when necessary*. There have been similar ideas outside the RL domains. For instance, Gu et al. (2016) and Satija & Pineau (2016) explore Real Time Neural Machine Translation where the action at every time step is to decide whether to output a new token in the target language or not based on current context.

Transition Point Dynamic Programming (TPDP) [Buckland & Lawrence (1994)] algorithm is a modification to the tabular dynamic programming paradigm that can reduce the learning time and memory required for control of continuous stochastic dynamic systems. This is done by determining a set of transition points in the underlying MDP. The policy changes only at these transition point states. The algorithm learns an optimal set of transition point states by using a variant of Q-Learning to evaluate whether or not to add/delete a particular state from the set of transition points. FiGAR learns the transition points in the underlying MDP on the fly with generalization across the state space unlike TPDP which is tabular and infeasible for large problems.

The Dynamic Frameskip Deep Q-network [Lakshminarayanan et al. (2017)] proposes to use multiple time scales of action repetition by augmenting the Deep Q Network (DQN) [Mnih et al. (2015)] with separate streams of the same primitive actions corresponding to each time scale. This way, the time scale of action repetition is dynamically learned. Although this framework leads to a significant improvement in the performance on a few Atari 2600 games, it suffers from not being able to support multiple time scales due to potential explosion of the action space and is restricted to discrete action spaces. Durugkar et al. (2016) also explore learning macro-actions composed using the same action repeated for different time scales. However, their framework is limited to discrete action spaces and performance improvements are not significant.

Learning temporally extended actions and abstractions have been of interest in RL for a long time. Vezhnevets et al. (2016) propose Strategic Attentive Writer (STRAW) for learning macro-actions and building dynamic action-plans directly from reinforcement learning signals. Instead of outputting a single action after each observation, STRAW maintains a multi-step action plan. The agent periodically updates the plan based on observations and commits to the plan between the replanning steps. Although the STRAW framework represents a more general temporal abstraction than FiGAR, FiGAR should be seen as a framework that can compliment STRAW whereby the decision to repeat could now be hierarchical at plan and base action levels.

FiGAR is a framework that has a structured policy representation where the time scale of execution could be thought as parameterizing the chosen action. The only other work that explores parameterized policies in DRL is Hausknecht & Stone (2016) where discrete actions are parameterized by continuous values. In our case, discrete/continuous actions are parameterized by discrete values. The state spaces in Atari are also more sophisticated than the kind explored in Hausknecht et al. (2016).

FiGAR is also very naturally connected to the Semi-MDPs (SMDPs) framework. SMDPs are MDPs with durative actions. The assumption in SMDPs is that actions take some *holding time* to complete [Duff (1995); Mahadevan et al. (1997); Dietterich (2000)]. Typically, they are modeled with two distributions, one corresponding to the next state transition and the other corresponding to the holding time which denotes the number of time steps between the current action from the policy until the next action from the policy. The rewards over the entire holding time of an action is the credit assigned for picking the action. In our framework, we naturally have durative actions due to the policy structure where the decision consists of both the choice of the action and the time scale of its execution. Therefore, we convert the original MDP to an SMDP trivially. In fact, we give more structure to the SMDP because we are clear that we *repeat* the chosen action during the holding time, while what happens during the holding time is not specified in the SMDP framework. One can think of the part of the policy that outputs the probability distribution over the time scales as a holding time distribution. Therefore, our framework naturally fits into the SMDP definition with the action repetition rate characterizing the holding time. We also sum up the rewards over the holding time with an an appropriate discounting factor as in an SMDP framework.

## 3 BACKGROUND

### 3.1 ASYNCHRONOUS ADVANTAGE ACTOR CRITIC

Actor critic algorithms execute policy gradient updates by maintaining parametric estimates for the policy $\pi_{\theta_a}(a|s)$ and the value function $V_{\theta_c}(s)$ [Sutton & Barto (1998)]. The value function estimates are used to reduce the variance in the policy gradient updates.

Asynchronous Advantage Actor Critic (A3C) [Mnih et al. (2016)] learns policies based on an asynchronous $n$-step returns. The $k$ learner threads execute $k$ copies of the policy asynchronously and the parameter updates are sent to a central parameter server at regular intervals. This ensures that temporal correlations are broken between subsequent updates since the different threads possibly explore different parts of the state space in parallel. The objective function for policy improvement in A3C is:

$$L(\theta_a) = \log \pi_{\theta_a}(a_t|s_t)\left(G_t - V(s_t)\right)$$

where $G_t$ is an estimate for the return at time step $t$. The A3C algorithm uses $n$-step returns for estimating $G_t$ which is a biased estimate for $Q(s_t, a_t)$. Hence one can think of $G_t - V(s_t)$ as an estimate for $A(s_t, a_t)$ which represents the advantage of taking action $a_t$ in state $s_t$. The value function $V_{\theta_c}(s_t)$ is updated by using $n$-step TD error as: $L(\theta_c) = \left(\hat{V}(s_t) - V_{\theta_c}(s_t)\right)^2$ where $\hat{V}(s_t)$ is an estimate of the $n$-step return from the current state. In A3C $j$-step returns are used where $j \leq n$ and $n$ is a fixed hyper-parameter. For simplicity assume that $t \leq n$. Then the definition for $\hat{V}(s_t)$ is:

$$\hat{V}(s_t) = \sum_{j=t}^{n-1} \gamma^{t-j} r_j + \gamma^{n-t} V(s_n)$$

The policy and value functions are parameterized by Deep Neural Networks.

## 3.2 Trust Region Policy Optimization

TRPO [Schulman et al. (2015)] is a policy optimization algorithm. Constrained optimization of a surrogate loss function is proposed, with theoretical guarantees for monotonic policy improvement. The TRPO surrogate loss function $L$ for potential next policies ($\tilde{\pi}$) is:

$$L_{\theta_{old}}(\tilde{\theta}) = \eta(\pi) + \sum_s \rho^{\pi}(s) \sum_a \tilde{\pi}(a|s) A_{\pi}(s, a)$$

where $\theta_{old}$ are the parameters of policy $\pi$ and $\tilde{\theta}$ are parameters of $\tilde{\pi}$. This surrogate loss function is optimized subject to the constraint:

$$D_{KL}^{\max}(\pi, \tilde{\pi}) \leq \delta$$

which ensures that the policy improvement can be done in non-trivial step sizes and at the same time the new policy does not deviate much from the current policy due to the KL-divergence constraint.

## 3.3 Deep Deterministic Policy Gradients

According to the Deterministic Policy Gradient (DPG) Theorem [Lever (2014)], the gradient of the performance objective ($J$) of the deterministic policy ($\mu$) in continuous action spaces with respect to the policy parameters ($\theta$) is given by:

$$\begin{aligned}
\nabla_\theta J(\mu_\theta) &= \int_{\mathbb{S}} \rho^\mu(s) \nabla_\theta \mu_\theta(s) \nabla_a Q^\mu(s, a)|_{a=\mu_\theta(s)} ds \\
&= \mathbb{E}_{s \sim \rho^\mu}[\nabla_\theta \mu_\theta(s) \nabla_a Q^\mu(s, a)|_{a=\mu_\theta(s)}]
\end{aligned} \tag{1}$$

for an appropriately defined performance objective $J$. The DPG model built according to this theorem consists of an actor which outputs an action vector in the continuous action space and a critic model $Q(s, a)$ which evaluates the action chosen at a state. The DDPG algorithm [Lillicrap et al. (2015)] extends the DPG algorithm by introducing non-linear neural network based function approximators for the actor and critic.

## 4 FiGAR: Fine Grained Action Repetition

FiGAR provides a DRL algorithm with the ability to model temporal abstractions by augmenting it with the ability to predict the number of time steps for which an action chosen for execution is to be repeated. This prediction is conditioned on the current state in the environment.

The FiGAR framework can be used to extend any DRL algorithm (say $Z$) which maintains an explicit policy. Let $Z'$ denote the extension of $Z$ under FiGAR. $Z'$ has two independent decoupled

---

**Algorithm 1** Create $FiGAR - Z$

---

1: **function** MAKEFIGAR(DRLAlgorithm Z, ActionRepetitionSet W)
2: $s_t \leftarrow$ state at time t
3: $a_t \leftarrow$ action taken in $s_t$ at time t
4: $\pi_a \leftarrow$ action policy of Z
5: $f_{\theta_a}(s_t) \leftarrow$ action network for realizing action policy $\pi_a$
6: $L(\pi_a, s_t, a_t) \leftarrow$ A's objective function for improving $\pi_a$
7: $\pi_x \leftarrow$ construct action repetition policy for FiGAR-Z.
8: $f_{\theta_x}(s_t) \leftarrow$ repetition network with output of size $|W|$ for action repetition policy $\pi_x$.
9: $L(\pi_x, s_t, a_t) \leftarrow$ L evaluated at $\pi_x$
10: $T(s_t, a_t) \leftarrow L(\pi_x, s_t, a_t) * L(\pi_a, s_t, a_t)$ // Total Loss
11: **return** T, $f_{\theta_a}$, $f_{\theta_x}$

---

policy components. The policy $\pi_{\theta_a}$ for choosing actions and the policy $\pi_{\theta_x}$ for choosing action repetitions. Algorithm 1 describes the generic framework for deriving DRL algorithm $Z'$ from algorithm Z. Let $W$ stand for the set of all action repetitions that $Z'$ would be able to perform. In tradition DRL algorithms, $W = \{c\}$, where $c$ is a constant. This implies that the action repetition is static and fixed. In FiGAR, The set of action repetitions from which $Z'$ can choose is $W = \{w_1, w_2, \cdots, w_{|W|}\}$. The central idea behind FiGAR is that the objective function used to update the parameters $\theta_a$ of $\pi_{\theta_a}$ maintained by Z will be used to update the parameters $\theta_x$ of the action repetition policy $\pi_{\theta_x}$ of $Z'$ as well (illustrated by the sharing of $L$ in Algorithm 1). In the first sub-section, we desribe how $Z'$ operates. In the next two sub-sections, we describe the instantiations of FiGAR extensions for 3 policy gradient DRL algorithms: A3C, TRPO and DDPG.

## 4.1 How FiGAR operates

The following procedure describes how FiGAR variant $Z'$ navigates the MDP that it is solving:

1. In the very first state $s_0$ seen by $Z'$, it predicts a tuple $(a_0, x_0)$ of action to execute and number of time steps for which to execute it. $a_0$ is decided based on $\pi_{\theta_a}(s_0)$ whereas $x_0$ is decided based on $\pi_{\theta_x}(s_0)$. Each such tuple is known as an action decision.

2. We denote by $s_j$ the state of the agent after $j$ such action decisions have been made. Similarly $x_j$ and $a_j$ denote the action repetition and the action chosen after $j$ such action decisions. Note that $x_j \in \{w_1, w_2, \cdots, w_{|W|}\}$, the set of all allowed action repetitions.

3. From time step 0 until $x_0$ , $Z'$ executes $a_0$.

4. At time step $x_0$, $Z'$ again decides, based on current state $s_1$ and policy components $(\pi_{\theta_a}(s_1), \pi_{\theta_x}(s_1))$, the tuple of action to execute and the number of times for which to execute it, $(a_1, x_1)$.

5. It can seen that in general if $Z'$ executes action $a_k$ for $x_k$ successive time steps, the next action is decided at time step $t = \sum_{i=0}^{k} x_i$ on the basis of $(\pi_{\theta_a}(s_{k+1}), \pi_{\theta_x}(s_{k+1}))$, where $s_{k+1}$ is the state seen at time step $t$.

## 4.2 FiGAR-A3C

A3C uses $f_{\theta_a}(s_j)$ and $f_{\theta_c}(s_j)$ which represent the policy $\pi(a|s_j)$ and the value function $V(s_j)$ respectively. $\pi(a|s_j)$ is a vector of size equal to the action space of the underlying MDP while $V(s_j)$ is a scalar. FiGAR extends the A3C algorithm as follows:

1. With $s_j$ defined as in the previous sub-section, in addition to $f_{\theta_a}(s_j)$ and $f_{\theta_c}(s_j)$ , FiGAR-A3C defines a neural network $f_{\theta_x}(s_j)$. This neural network outputs a $|W|$-dimensional vector representing the probability distribution over the elements of the set $W$. The sampled time scale from this multinomial distribution decides how long the action decided with $f_{\theta_a}(s_j)$ is repeated. The actor is now composed of both $f_{\theta_a}(s_j)$ (action network) and $f_{\theta_x}(s_j)$ (repetition network).

2. The objective function for the actor is modified to be:

$$L(\theta_a, \theta_x) = \left(\log f_{\theta_a}(a|s_j) + \log f_{\theta_x}(x|s_j)\right) A(s_j, a, x)$$

where $A(s_j, a, x)$ represents the advantage of executing action $a$ for $x$ time steps at state $s_j$. This implies that for FiGAR-A3C the combination operator $*$ defined in Algorithm 1 is in fact scalar addition.

3. The objective function for the critic is the same except that estimated value function used in the target for the critic is changed as:

$$\hat{V}(s_j) = \sum_{k=j}^{n-1} \gamma^{y_{k-j}} r_k + \gamma^{y_{n-j}} V(s_n)$$

where we define $y_0 = 0, y_k = y_{k-1} + x_k, k \geq 1$ and action $a_k$ was repeated $x_k$ times when state $s_k$ was encountered. Note that the return used in target is based on $n$ *decision steps*, steps at which a potential change in actions executed takes place. It is not based on $n$ time steps.

Note that point 2 above implies that the action space has been extended by $|W|$ and has a dimension of $|A| + |W|$. It is only because of this factored representation of the FiGAR policy that the number of parameters do not blow up. If one were to extend the action space in a naive way by coupling the actions and the action repetitions, one would end up suffering the kind of action-space blow-up as seen in [Lakshminarayanan et al. (2017); Vezhnevets et al. (2016)] wherein for being able to control with respect to $|W|$ different action repetition levels (or $|W|$-length policy plans in the case of STRAW), one would need to model $|A| \times |W|$ actions or action-values which would blow up the final layer size $|W|$ times.

## 4.3 FIGAR-TRPO

Although $f_{\theta_a}(s_j)$ in A3C is generic enough to output continuous or discrete actions, we consider A3C only for discrete action spaces. Preserving the notation from the previous subsection, we describe FiGAR-TRPO where we consider the case of the output generated by the network $f_{\theta_a}(s_j)$ to be $A$ dimensional with each dimension being independent and describing a continuous valued action. The stochastic policy is hence modeled as a multi-variate Gaussian with diagonal co-variance matrix. The parameters of the mean as well as the co-variance matrix are together represented by $\theta_a$ and the concatenated mean-covariance vector is represented by the function $f_{\theta_a}(s_j)$. FiGAR-TRPO is constructed as follows:

1. In TRPO, the objective function $L_{\theta_{old}}(\tilde{\theta})$ is constructed based on trajectories drawn according to the current policy. Hence, for FiGAR-TRPO the objective function is modified to be:

$$L_{\theta_{a,old}, \theta_{x,old}}(\tilde{\theta}_a) \times \left(L_{\theta_{a,old}, \theta_{x,old}}(\tilde{\theta}_x)\right)^{\beta_{ar}}$$

where $\theta_x$ are the parameters of sub-network $f_{\theta_x}$ which computes the action repetition distribution. This implies that for FiGAR-TRPO the combination operator $*$ defined in Algorithm 1 is in some sense the scalar multiplication. $\beta_{ar}$ controls the relative learning rate of the core-policy parameters and the action repetition parameters.

2. The constraint in TRPO corresponding to the KL divergence between old and new policies is modified to be:

$$D_{KL}^{\max}(\pi_a, \tilde{\pi}_a) + \beta_{KL} D_{KL}^{\max}(\pi_x, \tilde{\pi}_x) \leq \delta$$

where $\pi_a$ denotes the Gaussian distribution for the action to be executed and $\pi_x$ denotes the multinomial softmax-based action repetition probability distribution. $\beta_{KL}$ controls the relative divergence of $\pi_x$ and $\pi_a$ from the new corresponding policies. See Appendix $C$ for an explanation of the loss function used.

## 4.4 FIGAR-DDPG

In this subsection, we present an extension of DDPG under the FiGAR framework. DDPG consists of $f_{\theta_a}(s_j)$ which denotes a deterministic policy $\mu(s)$ and is a vector of size equal to the action space of the underlying MDP; and $f_{\theta_c}(s_j, a_j)$ which denotes the critic network whose output is a single number, the estimated state-action value function $Q(s_j, a_j)$. FiGAR framework extends the DDPG algorithm as follows:

1. $f_{\theta_x}$ is introduced, similar to FiGAR-A3C. This implies that the complete policy for FiGAR-DDPG $(\pi_{\theta_a}, \pi_{\theta_x})$ is computed by the tuple of neural networks: $(f_{\theta_a}, f_{\theta_x})$ . Similar to DDPG [Lillicrap et al. (2015)], FiGAR-DDPG has no loss function for the actor. The actor receives gradients from the critic. This is because the actors proposed policy is directly fed to the critic and the critic provides the actor with gradients which the proposed policy follows for improvement. In FiGAR-DDPG the total policy $\pi$ is a concatenation of vectors $\pi_a$ and $\pi_x$. Hence the gradients for the total policy are also simply the concatenation of the gradients for the policies $\pi_a$ and $\pi_x$.

2. To ensure sufficient exploration, the exploration policy for action repetition is an $\epsilon$-greedy version of the behavioral action repetition policy. The action part of the policy, $(f_{\theta_a}(s_j))$, continues to use temporally correlated noise for exploration, generated by an Ornstein-Uhlenbeck process (see Lillicrap et al. (2015) for details).

3. The critic is modeled by the equation

$$f(s_j, a_j, x_j) = f_{\theta_c}(s_j, f_{\theta_a}(s_j), f_{\theta_x}(s_j))$$

As stated above, $f_{\theta_x}$ is learnt by back-propagating the gradients produced by the critic with respect to $f_{\theta_x}$, in exactly the same way that $f_{\theta_a}$ is learnt.

## 5 EXPERIMENTAL SETUP AND RESULTS

The experiments are designed to understand the answers to the following questions:

1. For different DRL algorithms, can FiGAR extensions learn to use the dynamic action repetition?
2. How does FiGAR impact the performance of the different algorithms on various tasks?
3. Is FiGAR able to learn control on several different kinds of Action Repetition sets $W$?

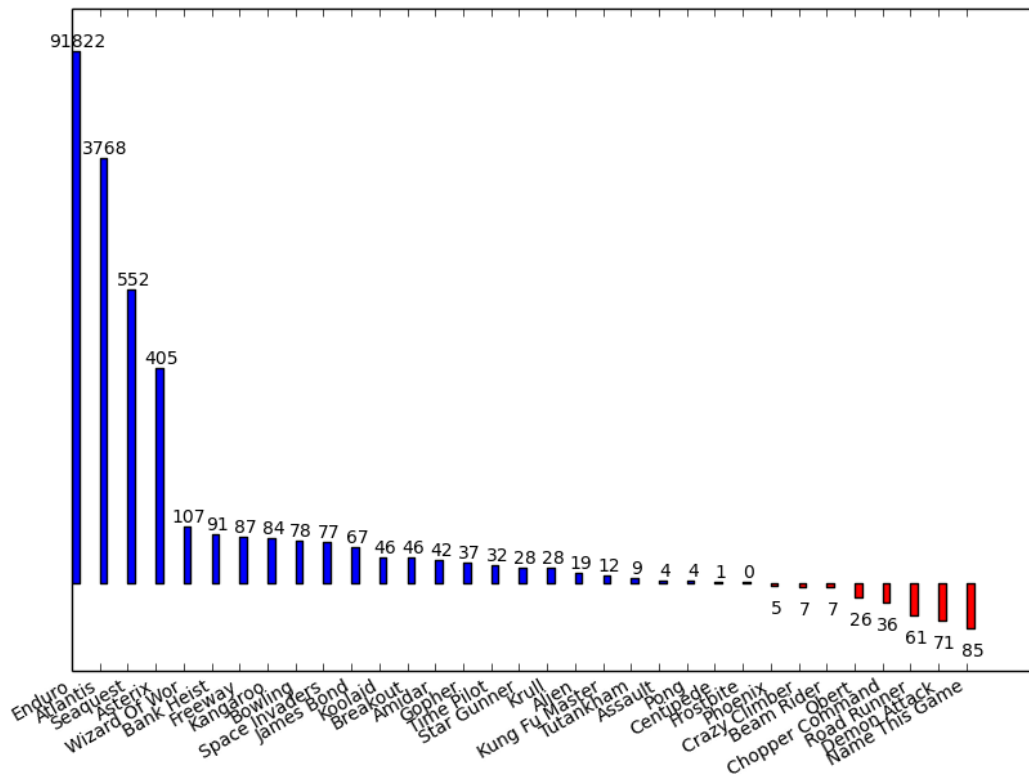

Figure 2: Percentage Improvement of FiGAR-A3C over A3C for Atari 2600

In the next three sub-sections, we experiment with the simplest possible action repetition set $W = \{1, 2, \cdots, |W|\}$. In the fourth sub-section, we understand the effects that changing the action repetition set $W$ has on the policies learnt.

## 5.1 FiGAR-A3C ON ATARI 2600

This set of experiments was performed with FiGAR-A3C on the Atari 2600 domain. The hyperparameters were tuned on a subset of games (Beamrider, Breakout, Pong, Seaquest and Space Invaders) and kept constant across all games.

$W$ is perhaps the most important hyper-parameter and depicts our confidence in the ability of a DRL agent to predict the future. Such a choice has to depend on the domain in which the DRL agent is operating. We only wanted to demonstrate the ability of FiGAR to learn temporal abstractions and hence instead of tuning for an optimal $|W|$, it was chosen to be 30, arbitrarily. The specific set of time scales we choose is $1, 2, 3, \cdots, 30$. FiGAR-A3C as well as A3C were trained for 100 million decision steps. They were evaluated in terms of the final policy learnt. Treating the score obtained by the A3C algorithm as baseline (b), we calculated the percentage improvement (i) offered by FiGAR-A3C (f) as: $i = \frac{f-b}{b}$. Figure 2 plots this metric versus the game names. The improvement for Enduro and Atlantis is staggering and more than $900\times$ and $35\times$ respectively. Figure 2's y-axis has been clipped at $1000\%$ to make it more presentable. Appendix $A$ contains the experimental details, the raw scores obtained by both the methods. Appendix $B$ contains experiments on validating our setup.

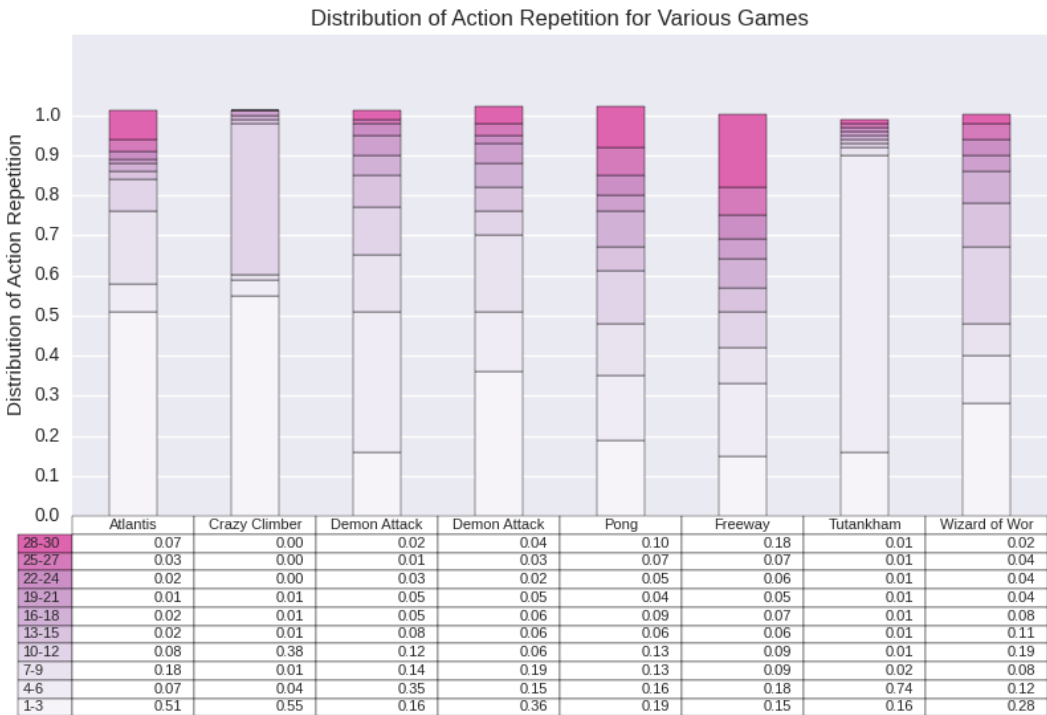

Figure 3: Evaluation of Action Repetition Control for Atari 2600. See Appendix $B$ (Table 7) for an expanded version of figure.

To answer the first question we posed, experiments were conducted to record the percentage of times that a particular action repetition was chosen. Figure 3 presents the action repetition distribution across a selection of games, chosen arbitrarily. The values have been rounded to 2 decimal places and hence do not sum to 1 in each game. Each game was played for 10 episodes using the same policy used to calculate average scores in Figure 2.
The two tables together show that FiGAR-A3C generally prefers lower action repetition but does come up with temporal abstractions in policy space (specially in games like Pong and Crazy Climber). Some such abstractions have been demonstrated in Figure 1. Such temporal abstractions do not always help general gameplay (Demon Attack). However, as can be seen from Figure 2, FiGAR-A3C outperforms A3C in 26 out of 33 games.
One could potentially think of FiGAR as a deep exploration framework by using the learnt policy $\pi_{\theta_a}$ for predicting actions at every time step and completely discarding the action-repetition policy

$\pi_{\theta_x}$, at evaluation time. Appendix $F$ contains an empirical argument against such a usage of FiGAR and demonstrates that the temporal abstractions encoded by $\pi_{\theta_x}$ are indeed important for game play performance.

## 5.2 FiGAR-TRPO on Mujoco Tasks

In this sub-section we demonstrate that FiGAR-TRPO can learn to solve the Mujoco simulated physics tasks reasonably successfully. Similar to FiGAR-A3C, $|W|$ is chosen to be 30 arbitrarily.

Table 1: Evaluation of FiGAR on Mujoco

| Domain | FiGAR-TRPO | TRPO |
|---|---|---|
| Ant | **947.06** (28.35) | -161.93 (1.00) |
| Hopper | 3038.63 (1.00) | **3397.58** (1.00) |
| Inverted Pendulum | **1000.00** (1.00) | 971.66 (1.00) |
| Inverted Double Pendulum | **8712.46** (1.01) | 8327.75 (1.00) |
| Swimmer | 337.48 (10.51) | **364.55** (1.00) |

The full policy $(f_{\theta_a}, f_{\theta_x})$ is trained jointly. The policies learnt after each TRPO optimization step (details in Appendix $C$) are compared to current best known policy to arrive at the overall best policy. The results in this sub-section are for this best policy. Table 1 compares the performance of TRPO and FiGAR-TRPO. The number in the brackets is the average action repetition chosen. As can be seen from the table, FiGAR learns either policies which are much faster to execute albeit at cost of slight loss in optimality or it learns policies similar to non-repetition case, performance being competitive with the baseline algorithm. This best policy was then evaluated on 100 episodes to arrive at average scores which are contained in Table 1. TRPO is a difficult baseline on the MuJoCo tasks domain. On the whole, FiGAR outperforms TRPO in 3 out of 5 domains, although the gains are marginal in most tasks. Appendix $C$ contains experimental details. A video showing FiGAR-TRPO's learned behavior policies can be found at `http://youtu.be/JiaO2tBtH-k`.

## 5.3 FiGAR-DDPG on Torcs

FiGAR-DDPG was trained and tested on the TORCS domain. $|W|$ was chosen to be 15 arbitrarily. FIGAR-DDPG manages to complete the race task flawlessly and manages to finish 20 laps of the circuit, after which the simulator stops. The total reward obtained by FiGAR-DDPG was 557929.68 as against 59519.70 obtained by DDPG. We also observed that FiGAR-DDPG learnt policies which were smoother than those learnt by DDPG. A video showing the learned driving behavior of the FiGAR-DDPG agent can be found at `https://youtu.be/dX8J-sF-WX4`. See Appendix $D$ for experimental and architectural details.

## 5.4 Effect of Action Repetition Set on FiGAR

This sub-section answers the third question raised at the beginning of this section in affirmative. We demonstrate that there is nothing sacrosanct about the set of action repetitions $W = \{1, 2, \cdots, 30\}$ on which FiGAR-A3C performed well, and that the good performance carries over to other action repetition sets.

To demonstrate the generality of FiGAR with respect to $W$, we chose a wide variety of action repetition sets $W$, trained and evaluated FiGAR-A3C variants which learn to repeat with respect to their respective Action Repetition sets. Table 3 describes the various FiGAR-variants considered for these experiments in terms of their action repetition set $W$.

Note that the hyper-parameters of the various variants of FiGAR-A3C were not tuned but rather the same ones obtained by tuning for FiGAR-30 were used. Table 2 contains a comparison of the raw scores obtained by the various FiGAR-A3C variants in comparison to the A3C baseline. It is clear that FiGAR is able to learn over any action repetition set $W$ and the performance does not fall by a lot even when hyper-parameters tuned for FiGAR-30 are used for other variants. Appendix $E$

Table 2: Comparison of FiGAR-A3C variants to the A3C baseline for 3 games: Sea Quest, Space Invaders and Asterix. See Appendix E (Figure 7) for a bar graph visualization of this table.

| Variant | Seaquest | Space Invaders | Asterix |
|---|---|---|---|
| FiGAR-50 | **22904.50** | 1929.50 | 7730.00 |
| FiGAR-30-50 | 17103.60 | 1828.90 | 11090.00 |
| FiGAR-P | 20005.40 | 2047.40 | 10937.00 |
| FiGAR-30 | 18076.90 | 2251.95 | **11949.00** |
| FiGAR-20-30 | 14683.00 | **2310.70** | 8182.00 |
| FiGAR-20 | 19148.50 | 1929.50 | 7730.00 |
| Baseline | 2769.40 | 1268.75 | 2364.00 |

Table 3: Description of FiGAR-A3C variants in terms of action repetition set $W$.

| **Name** | Description in terms of $W$ |
|---|---|
| FiGAR-20 | $W = \{1, 2, \cdots, 19, 20\}$ |
| FiGAR-30 | $W = \{1, 2, \cdots, 29, 30\}$ |
| FiGAR-50 | $W = \{1, 2, \cdots, 49, 50\}$ |
| FiGAR-30-50 | $W = \{30 \text{ numbers drawn randomly from } W = \{1, 2, \cdots, 50\} \text{ w/o replacement}\}$ |
| FiGAR-20-30 | $W = \{20 \text{ numbers drawn randomly from } W = \{1, 2, \cdots, 30\} \text{ w/o replacement}\}$ |
| FiGAR-P | $W = \{p \mid p < 50, \ p \in \mathbb{P} \text{ (Set of all Primes)}\}$ |

contains additional graphs showing the evolution of average game scores against number of training steps as well as a bar graph visualization of Table 2.

# 6 CONCLUSION, SHORTCOMINGS AND FUTURE WORK

We propose a light-weight framework (FiGAR) for improving current Deep Reinforcement Learning algorithms for policy optimization whereby temporal abstractions are learned in the policy space. The framework is generic and applicable to DRL algorithms concerned with policy gradients for continuous as well as discrete action spaces such as A3C, TRPO and DDPG. FiGAR maintains a structured policy wherein the action probability distribution is augmented with a probability distribution for choosing the time scale of repeating the chosen action. Our results demonstrate that FiGAR can be used to significantly improve the current policy gradient and Actor-Critic algorithms thereby learning better control policies across several domains by discovering optimal sequences of temporally elongated macro-actions.

Atari, TORCS and MuJoCo represent environments which are largely deterministic with a minimal degree of stochasticity in environment dynamics. In such highly deterministic environments we would expect FiGAR agents to build a latent model of the environment dynamics and hence be able to execute large action repetitions without dying. This is exactly what we see in a highly deterministic environment like the game Freeway. Figure 1 (a) demonstrates that the chicken is able to judge the speed of the approaching cars appropriately and cross the road in a manner which takes it to the goal without colliding with the cars and at the same time avoiding them narrowly.

Having said that, certainly the ability to stop an action repetition (or a macro-action) in general would be very important, especially in stochastic environments. In our setup, we do not consider the ability to stop executing a macro-action that the agent has committed to. However, this is a necessary skill in the event of unexpected changes in the environment while executing a chosen macro-action. Thus, stop and start actions for stopping and committing to macro-actions can be added to the basic dynamic time scale setup for more robust policies. We believe the modification could work for more general stochastic worlds like Minecraft and leave it for future work.

## ACKNOWLEDGMENTS

We used the open source implementation of A3C at `https://github.com/miyosuda/async_deep_reinforce`. We thank Volodymr Mnih for giving valuable hyper-parameter information. We thank Aravind Rajeswaran (University of Washington) for very helpful discussions regarding and feedback on the MuJoCo domain tasks. The TRPO implementation was a modification of `https://github.com/aravindr93/robustRL`. The DDPG implementation was a modification of `https://github.com/yanpanlau/DDPG-Keras-Torcs`. We thank ILDS (`http://web.iitm.ac.in/ilds/`) for the compute resources we used for running A3C experiments.

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

## APPENDIX A: EXPERIMENTAL DETAILS FOR FiGAR-A3C

### EXPERIMENTAL DETAILS AND RESULTS

We used the LSTM-variant of A3C [Mnih et al. (2016)] algorithm for FiGAR-A3C experiments. The async-rmsprop algorithm [Mnih et al. (2016)] was used for updating parameters with the same hyper-parameters as in Mnih et al. (2016). The initial learning rate used was $10^{-3}$ and it was linearly annealed to 0 over 100 million steps. The $n$ used in $n$-step returns was 20. Entropy regularization was used to encourage exploration, similar to Mnih et al. (2016). The $\beta$ for entropy regularization was found to be 0.02 after hyper-parameter tuning, both for the action-policy $f_{\theta_a}$ and the action repetition policy $f_{\theta_x}$.

Table 4: Game Playing Experiments on Atari 2600

| Name | FiGAR-A3C | A3C |
|---|---|---|
| Alien | **3138.50** (2864.91, 3412.08) | 2709.20 (2499.41, 2918.98) |
| Amidar | **1465.70** (1406.18, 1525.21) | 1028.34 (1003.11, 1053.56) |
| Assault | **1936.37** (1855.85, 2016.88) | 1857.61 (1787.19, 1928.02) |
| Asterix | **11949.00** (11095.62, 12802.37) | 2364.00 (2188.12, 2539.87) |
| Atlantis | **6330600.00** (6330600.00, 6330600.00) | 163660.00 (-46665.38, 373985.38) |
| Bank Heist | **3364.60** (3342.10, 3387.09) | 1731.40 (1727.94, 1734.85) |
| Beam Rider | **2348.78** (2152.19, 2545.36) | 2189.96 (2062.89, 2317.02) |
| Bowling | **30.09** (29.74, 30.43) | 16.88 (15.23, 18.52) |
| Breakout | **814.50** (789.97, 839.02) | 555.05 (474.89, 635.20) |
| Centipede | **3340.35** (3071.70, 3608.99) | 3293.33 (2973.14, 3613.51) |
| Chopper command | 3147.00 (2851.02, 3442.97) | **4969.00** (4513.12, 5424.87) |
| Crazy Climber | 154177.00 (148042.35, 160311.64) | **166875.00** (161560.18, 172189.81) |
| Demon Attack | 7499.30 (7127.85, 7870.74) | **26742.75** (22665.02, 30820.47) |
| Enduro | **707.80** (599.16, 816.43) | 0.77 (0.45, 1.09) |
| Freeway | **33.14** (33.01, 33.26) | 17.68 (17.41, 17.94) |
| Frostbite | **309.60** (308.81, 310.38) | 306.80 (304.67, 308.92) |
| Gopher | **12845.40** (11641.88, 14048.91) | 9360.60 (8683.72, 10037.47) |
| James Bond | **478.0** (448.78, 507.21) | 285.5 (268.62, 302.37) |
| Kangaroo | **48.00** (29.51, 66.48) | 26.00 (12.81, 39.18) |
| Koolaid | **1669.00** (1583.58, 1754.42) | 1136.0 (1065.36, 1206.64) |
| Krull | **1316.10** (1223.23, 1408.96) | 1025.00 (970.77, 1079.22) |
| Kung Fu Master | **40284.00** ( 38207.21, 42360.78) | 35717.00 (34288.21, 37145.78) |
| Name this game | 1752.60 (1635.77, 1869.42) | **12100.80** (11682.64, 12518.95) |
| Phoenix | 5106.10 (5056.43, 5155.76) | **5384.10** (5178.12, 5590.07) |
| Pong | **20.32** (20.17, 20.46) | 19.46 (19.32, 19.59) |
| Q-bert | 18922.50 (17302.94, 20542.05) | **25840.25** (25528.49, 26152.00) |
| Road Runner | 22907.00 ( 22283.32, 23530.67) | **59540.00** (58835.01, 60244.98) |
| Sea quest | **18076.90** (16964.16, 19189.63) | 2799.60 (2790.22, 2808.97) |
| Space Invaders | **2251.95** (2147.13, 2356.76) | 1268.75 (1179.25, 1358.24) |
| Star Gunner | **51269.00** (48629.42, 53908.57) | 39835.00 (36365.24, 43304.75) |
| Time Pilot | **11865.00** (11435.25, 12294.74) | 8969.00 (8595.57, 9342.42) |
| Tutankhamun | **276.95** (274.22, 279.67) | 252.82(241.38, 264.25) |
| Wizard of Wor | **6688.00** (5783.48, 7592.51) | 3230.00 (2355.75, 4104.24) |

Since the Atari 2600 games tend to be quite complex, jointly learning a factored policy from random weight initializations proved to be less optimal as compared to a more stage-wise approach. The approach we followed for training FiGAR-A3C was to first train the networks using the regular A3C-objective function. This stage trains the action part of the policy $f_{\theta_a}$ and value function $f_{\theta_c}$ for a small number of iterations with a fixed action repetition rate (in this stage, gradients are not back-propagated for $f_{\theta_x}$ and all action repetition predictions made are discarded). The next stage was to then train the entire architecture $(f_{\theta_a}, f_{\theta_x}, f_{\theta_c})$ jointly. This kind of a non-stationary training objective ensures that we have a good value function estimator $f_{\theta_c}$ and a good action policy

estimator $f_{\theta_a}$ before we start training the full policy $(f_{\theta_a}, f_{\theta_x})$ jointly. Every time FiGAR decides to execute action $a_t$ for $x_t$ time steps, we say one step of action selection has been made. Since the number of time steps for which an action is repeated is variable, training time is measured in terms of action selections carried out. The first stage of the training was executed for 20 million (a hyper-parameter we found by doing grid search) action selections (called steps here onwards) and the next stage was executed for 80 million steps. In comparison the baseline ran for 100 million steps (action selections).

Since a large entropy regularization was required to explore both components ($f_a$ and $f_x$) of the policy-space, this also ends up meaning that the policies learnt are more diffused than one would like them to be. Evaluation was done after every 1 million steps and followed a strategy similar to $\epsilon$-greedy. With $\epsilon = 0.1$ probability, the action and action repetition was drawn from the output distribution (($f_{\theta_a}$ and $f_{\theta_x}$ respectively) and with probability $1 - \epsilon$ the action (and independently the action selection) with maximum probability was selected. This evaluation was done for 100 episodes or 100000 steps whichever was smaller, to arrive at an average score.

Table 4 contains the raw scores obtained by the final FiGAR-A3C and A3C policies on 33 Atari 2600 games. The numbers inside the brackets depict the confidence interval at a confidence threshold of 0.95, calculated by averaging scores over 100 episodes. Table 5 contains scores for a competing method, STRAW [Vezhnevets et al. (2016)], which learns temporal abstractions by maintaining action plans, for the subset of games on which both FiGAR and STRAW were trained and tested. Note that the scores obtained by STRAW agents are averages over top 5 performing replicas. We can infer from Tables 4 and 5 that FiGAR and STRAW and competitive with each other, with FiGAR clearly out-performing STRAW in Breakout and STRAW clearing outperforming FiGAR in Frostbite.

Table 5: Game Playing Experiments on Atari 2600 by STRAW [Vezhnevets et al. (2016)]

| Name | STRAW | STRAW-e |
|---|---|---|
| Alien | 2626 | **3230** |
| Amidar | **2223** | 2022 |
| Breakout | 344 | **386** |
| Crazy Climber | 143803 | **153327** |
| Frostbite | 4394 | **8108** |
| Q-bert | 20933 | **23892** |

Figure 4 demonstrates the evolution of the performance of FiGAR-A3C versus training progress. It also contains corresponding metrics for A3C to facilitate comparisons. In the 100 episode long evaluation phase we also keep track of the best episodic score. We also plot the best episode's score versus time to get an idea of how bad the learnt policy is compared to the best it could have been.

ARCHITECTURE DETAILS

We used the same low level architecture as Mnih et al. (2016) which in turn uses the same low level architecture as Mnih et al. (2015), except that the pre-LSTM hidden layer had size 256 instead of 512 as in Mnih et al. (2016). Similar to Mnih et al. (2016) the Actor and Critic share all but one layer. Hence all but the final layer of $f_{\theta_a}$, $f_{\theta_x}$ and $f_{\theta_c}$ are the same. Each of the 3 networks has a different final layer with $f_{\theta_a}$ and $f_{\theta_x}$ having a softmax-non linearity as output non-linearity, to model the multinomial distribution and the $f_{\theta_c}$ (critic)'s output being linear.

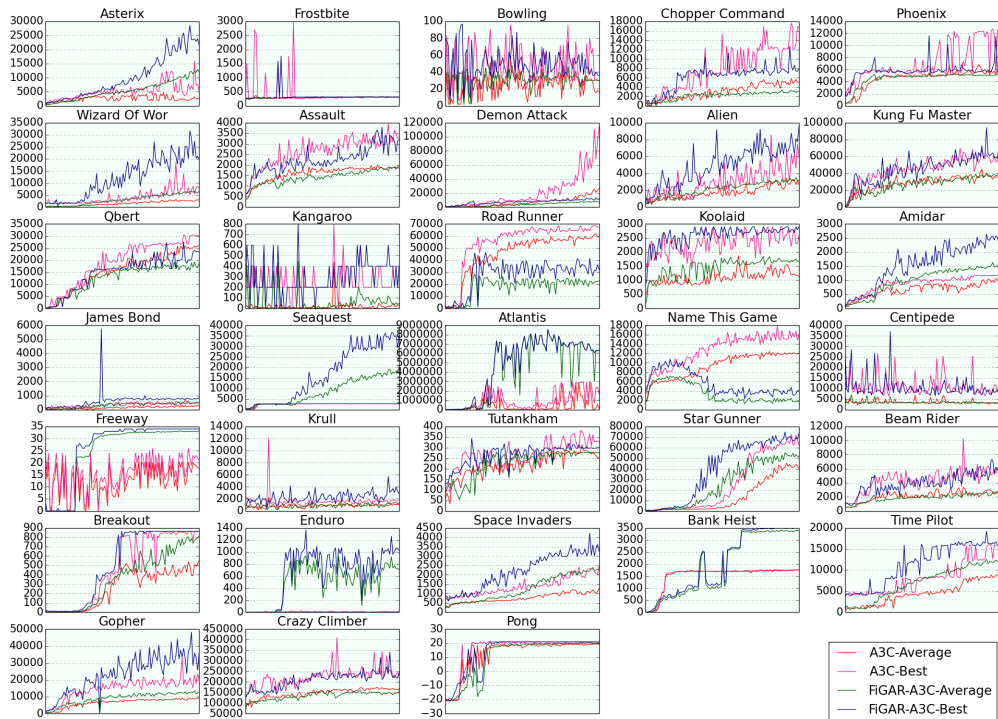

Figure 4: Training progress plotted versus time for Atari 2600

## APPENDIX B: ADDITIONAL EXPERIMENTS FOR ATARI 2600

These additional experiments are geared at understanding the repercussions of the evaluation strategy chosen by us.

### THE CHOICE OF WHETHER TO BE GREEDY OR STOCHASTIC

Note that in Appendix $A$, we state that for evaluating the policy learnt by the agent, we simply chose to sample from the output probability distributions with probability $0.1$ and chose the optimal action/action repetition with probability $0.9$. This choice of $0.1$ might seem rather arbitrary. Hence we conducted experiments to understand how well the agent performs as we shift more and more from choosing the maximal action($0.1$-greedy policy) towards sampling from output distributions (stochastic policy).

Figure 5 demonstrates that the performance of FiGAR-A3C does not deteriorate significantly, in comparison to A3C, even if we always sample from policy distributions, for most of the games. In the cases that there is a significant deterioration, we believe it is due to the diffused nature of the policy distributions (action and action repetition) learnt. Hence, although our choice of evaluation scheme might seem arbitrary, it is in fact reasonable.

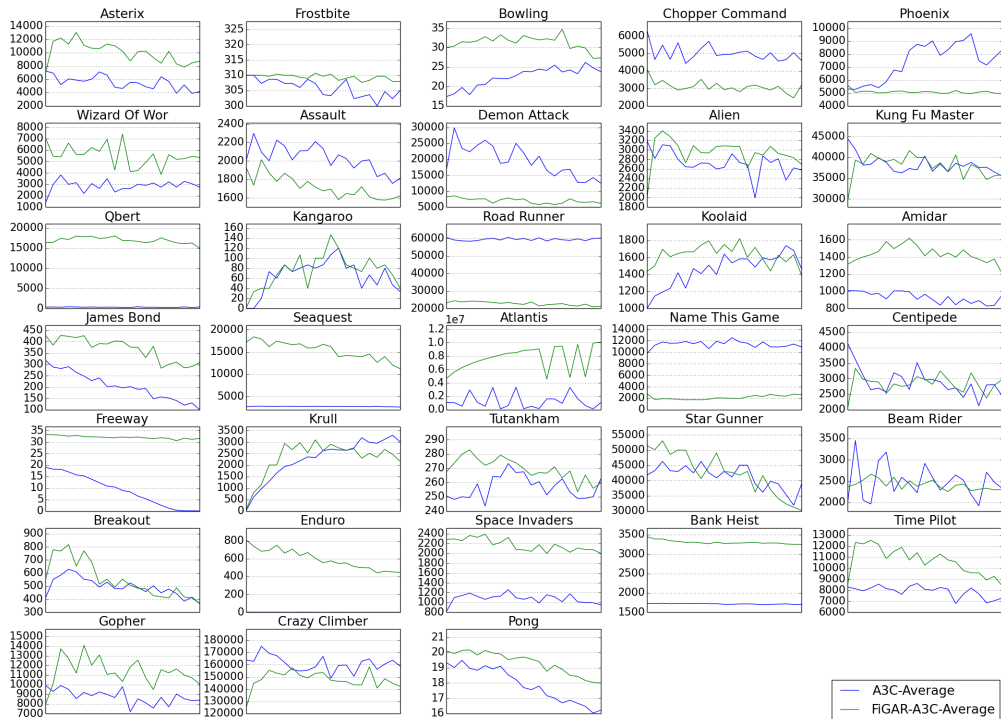

Figure 5: Average performance plotted against the probability with which we sample from final policy distribution for Atari 2600. Points toward the left side of a sub-graph depict average performance for a greedy version of a policy and those towards the right side depict performance for the stochastic version of the policy.

PERFORMANCE VERSUS SPEED TRADEOFF

The previous discussion leads to a novel way to trade-off game-play performance versus speed. Figure 3 demonstrated that although FiGAR-A3C learns to use temporally elongated macro-actions, it does favor shorter actions for many games. Since the action repetition distribution $\pi_{\theta_x}$ is diffused (as will be shown by Table 6), sampling from the distribution should help FiGAR choose larger action repetition rates probably at the cost of optimality of game play.

Table 6 demonstrates that this is exactly what FiGAR does. It was generated by playing 10 episodes, or 100000 steps, whichever is lesser and recording the fraction of times each action repetition was chosen. The policy used in populating table 6 was the stochastic policy (described in previous subsection). Contrast Table 6 to Table 7 which is an expanded version of Figure 3.

Table 6: Distribution of Action Repetitions chosen when the policy (both $\pi_{\theta_a}$ and $\pi_{\theta_x}$) is completely stochastic

| **Name** | 1-3 | 4-6 | 7-9 | 10-12 | 13-15 | 16-18 | 19-21 | 22-24 | 25-27 | 28-30 |
|---|---|---|---|---|---|---|---|---|---|---|
| Alien | 0.33 | 0.15 | 0.13 | 0.11 | 0.13 | 0.07 | 0.03 | 0.02 | 0.014 | 0.01 |
| Amidar | 0.19 | 0.14 | 0.10 | 0.08 | 0.08 | 0.07 | 0.06 | 0.08 | 0.09 | 0.12 |
| Assault | 0.29 | 0.26 | 0.21 | 0.11 | 0.04 | 0.03 | 0.02 | 0.01 | 0.01 | 0.01 |
| Asterix | 0.40 | 0.25 | 0.15 | 0.08 | 0.04 | 0.04 | 0.02 | 0.02 | 0.01 | 0.01 |
| Atlantis | 0.25 | 0.16 | 0.11 | 0.09 | 0.08 | 0.06 | 0.05 | 0.07 | 0.06 | 0.08 |
| Bank Heist | 0.950 | 0.04 | 0.00 | 0.00 | 0.00 | 0.00 | 0.00 | 0.00 | 0.00 | 0.00 |
| Beam Rider | 0.17 | 0.16 | 0.14 | 0.11 | 0.09 | 0.07 | 0.06 | 0.05 | 0.06 | 0.09 |
| Bowling | 0.01 | 0.01 | 0.01 | 0.01 | 0.01 | 0.01 | 0.01 | 0.01 | 0.01 | 0.91 |
| Breakout | 0.28 | 0.20 | 0.13 | 0.09 | 0.06 | 0.05 | 0.04 | 0.05 | 0.04 | 0.07 |
| Centipede | 0.19 | 0.27 | 0.34 | 0.17 | 0.03 | 0.00 | 0.00 | 0.00 | 0.00 | 0.00 |
| Chpr Cmd | 0.12 | 0.14 | 0.11 | 0.08 | 0.11 | 0.12 | 0.10 | 0.08 | 0.08 | 0.06 |
| Crzy Clmbr | 0.34 | 0.06 | 0.03 | 0.51 | 0.02 | 0.01 | 0.01 | 0.01 | 0.01 | 0.01 |
| Dmn Attk | 0.18 | 0.21 | 0.16 | 0.13 | 0.10 | 0.08 | 0.06 | 0.04 | 0.03 | 0.02 |
| Enduro | 0.66 | 0.34 | 0.00 | 0.00 | 0.00 | 0.00 | 0.00 | 0.00 | 0.00 | 0.00 |
| Pong | 0.16 | 0.15 | 0.13 | 0.10 | 0.10 | 0.08 | 0.08 | 0.07 | 0.08 | 0.07 |
| Freeway | 0.14 | 0.12 | 0.11 | 0.10 | 0.09 | 0.09 | 0.08 | 0.08 | 0.08 | 0.12 |
| Frostbite | 0.33 | 0.16 | 0.08 | 0.07 | 0.05 | 0.03 | 0.03 | 0.02 | 0.07 | 0.14 |
| Gopher | 0.41 | 0.15 | 0.23 | 0.07 | 0.04 | 0.03 | 0.02 | 0.02 | 0.01 | 0.01 |
| James Bond | 0.12 | 0.11 | 0.10 | 0.10 | 0.10 | 0.09 | 0.11 | 0.09 | 0.09 | 0.10 |
| Kangaroo | 0.10 | 0.10 | 0.11 | 0.10 | 0.11 | 0.10 | 0.10 | 0.10 | 0.09 | 0.09 |
| Koolaid | 0.14 | 0.14 | 0.11 | 0.11 | 0.10 | 0.08 | 0.08 | 0.09 | 0.08 | 0.07 |
| Krull | 0.92 | 0.01 | 0.01 | 0.01 | 0.01 | 0.01 | 0.01 | 0.01 | 0.00 | 0.00 |
| Kung Fu | 0.32 | 0.15 | 0.10 | 0.10 | 0.08 | 0.06 | 0.05 | 0.05 | 0.05 | 0.04 |
| NTG | 0.10 | 0.10 | 0.12 | 0.11 | 0.10 | 0.11 | 0.09 | 0.10 | 0.09 | 0.09 |
| Phoenix | 0.32 | 0.15 | 0.11 | 0.07 | 0.06 | 0.05 | 0.06 | 0.06 | 0.07 | 0.05 |
| Pong | 0.15 | 0.15 | 0.14 | 0.10 | 0.09 | 0.08 | 0.07 | 0.07 | 0.07 | 0.08 |
| Q-bert | 0.40 | 0.30 | 0.06 | 0.03 | 0.02 | 0.02 | 0.01 | 0.01 | 0.01 | 0.14 |
| Road Runner | 0.99 | 0.00 | 0.00 | 0.00 | 0.01 | 0.00 | 0.00 | 0.00 | 0.00 | 0.00 |
| Sea Quest | 0.40 | 0.26 | 0.10 | 0.05 | 0.04 | 0.04 | 0.04 | 0.03 | 0.02 | 0.01 |
| Spc Invdr | 0.33 | 0.16 | 0.11 | 0.07 | 0.06 | 0.05 | 0.04 | 0.04 | 0.06 | 0.09 |
| Star Gunner | 0.42 | 0.31 | 0.14 | 0.06 | 0.03 | 0.01 | 0.01 | 0.01 | 0.01 | 0.00 |
| Time Pilot | 0.14 | 0.16 | 0.15 | 0.12 | 0.09 | 0.07 | 0.07 | 0.06 | 0.06 | 0.08 |
| Tutankham | 0.34 | 0.18 | 0.08 | 0.08 | 0.07 | 0.06 | 0.06 | 0.05 | 0.05 | 0.04 |
| Wzd of Wor | 0.11 | 0.11 | 0.11 | 0.11 | 0.12 | 0.11 | 0.09 | 0.09 | 0.08 | 0.07 |

Both Figure 3 and Table 7 were created using the $0.1$-greedy policy described in previous subsection. The reason that we compare the stochastic policy with the $0.1$-greedy version instead of the fully-greedy version (wherein the optimal action and action repetition is always chosen) is that such a policy would end up being deterministic would not be good for evaluations.
It can hence be seen that FiGAR learns to trade-off optimality of game-play for speed by choosing whether to sample from policy probability distributions ($\pi_{\theta_a}$ and $\pi_{\theta_x}$) with probability 1 and thus behave stochastically, or behave $0.1$-greedily, and sample from the distributions with only a small probability. Table 6 can be compared to Figure 3 to understand how stochasticity in final policy affects action repetition chosen. A clear trend can be seen in all games wherein the stochastic variant of final policy learns to use longer and longer actions, albeit at a small cost of some loss in the optimality of game-play (as shown by Figure 5).

An expanded version of Figure 3 is presented as Table 7 for comparison with Table 6. As explained in Appendix $A$, the policy used for populating Table 7 is such that it picks a greedy action (or action repetition) with probability $0.9$ and stochastically samples from output probability distributions with probability $0.1$.

Table 7: Distribution of Action Repetitions chosen when the policy (both $\pi_{\theta_a}$ and $\pi_{\theta_x}$) is 0.1-greedy

| Name | 1-3 | 4-6 | 7-9 | 10-12 | 13-15 | 16-18 | 19-21 | 22-24 | 25-27 | 28-30 |
|---|---|---|---|---|---|---|---|---|---|---|
| Alien | 0.50 | 0.08 | 0.11 | 0.07 | 0.12 | 0.07 | 0.02 | 0.02 | 0.01 | 0.01 |
| Amidar | 0.49 | 0.08 | 0.06 | 0.04 | 0.04 | 0.04 | 0.04 | 0.07 | 0.03 | 0.11 |
| Assault | 0.45 | 0.26 | 0.15 | 0.06 | 0.02 | 0.02 | 0.02 | 0.01 | 0.01 | 0.01 |
| Asterix | 0.50 | 0.33 | 0.09 | 0.04 | 0.01 | 0.01 | 0.01 | 0.00 | 0.00 | 0.00 |
| Atlantis | 0.51 | 0.07 | 0.18 | 0.08 | 0.02 | 0.02 | 0.01 | 0.02 | 0.03 | 0.07 |
| Bank Heist | 0.96 | 0.04 | 0.00 | 0.00 | 0.00 | 0.00 | 0.00 | 0.00 | 0.00 | 0.00 |
| Beam Rider | 0.34 | 0.31 | 0.13 | 0.04 | 0.05 | 0.03 | 0.01 | 0.02 | 0.02 | 0.06 |
| Bowling | 0.01 | 0.91 | 0.01 | 0.01 | 0.01 | 0.01 | 0.01 | 0.01 | 0.01 | 0.01 |
| Breakout | 0.29 | 0.23 | 0.12 | 0.09 | 0.05 | 0.04 | 0.02 | 0.03 | 0.03 | 0.11 |
| Centipede | 0.02 | 0.03 | 0.94 | 0.02 | 0.00 | 0.00 | 0.00 | 0.00 | 0.00 | 0.00 |
| Chpr Cmd | 0.29 | 0.23 | 0.12 | 0.03 | 0.06 | 0.09 | 0.06 | 0.04 | 0.06 | 0.03 |
| Crzy Clmbr | 0.55 | 0.04 | 0.01 | 0.38 | 0.01 | 0.01 | 0.01 | 0.00 | 0.00 | 0.00 |
| Dmn Attk | 0.16 | 0.35 | 0.14 | 0.12 | 0.08 | 0.05 | 0.05 | 0.03 | 0.01 | 0.02 |
| Enduro | 0.91 | 0.09 | 0.00 | 0.00 | 0.00 | 0.00 | 0.00 | 0.00 | 0.00 | 0.00 |
| Freeway | 0.15 | 0.18 | 0.09 | 0.09 | 0.06 | 0.07 | 0.05 | 0.06 | 0.07 | 0.18 |
| Frostbite | 0.47 | 0.20 | 0.13 | 0.01 | 0.03 | 0.01 | 0.01 | 0.00 | 0.03 | 0.11 |
| Gopher | 0.47 | 0.19 | 0.21 | 0.05 | 0.04 | 0.01 | 0.02 | 0.01 | 0.00 | 0.00 |
| James Bond | 0.28 | 0.11 | 0.22 | 0.08 | 0.06 | 0.06 | 0.05 | 0.05 | 0.03 | 0.06 |
| Kangaroo | 0.20 | 0.39 | 0.27 | 0.02 | 0.01 | 0.04 | 0.01 | 0.01 | 0.01 | 0.04 |
| Koolaid | 0.36 | 0.15 | 0.19 | 0.06 | 0.06 | 0.06 | 0.05 | 0.02 | 0.03 | 0.04 |
| Krull | 0.92 | 0.01 | 0.01 | 0.01 | 0.01 | 0.01 | 0.01 | 0.01 | 0.00 | 0.00 |
| Kung Fu | 0.46 | 0.10 | 0.05 | 0.11 | 0.08 | 0.06 | 0.04 | 0.03 | 0.04 | 0.05 |
| NTG | 0.01 | 0.01 | 0.91 | 0.01 | 0.01 | 0.01 | 0.01 | 0.01 | 0.01 | 0.01 |
| Phoenix | 0.44 | 0.44 | 0.04 | 0.02 | 0.01 | 0.01 | 0.01 | 0.01 | 0.02 | 0.01 |
| Pong | 0.19 | 0.16 | 0.13 | 0.13 | 0.06 | 0.09 | 0.04 | 0.05 | 0.07 | 0.10 |
| Q-bert | 0.51 | 0.27 | 0.05 | 0.02 | 0.00 | 0.00 | 0.00 | 0.00 | 0.01 | 0.13 |
| Road Runner | 1.00 | 0.00 | 0.00 | 0.00 | 0.00 | 0.00 | 0.00 | 0.00 | 0.00 | 0.00 |
| Sea Quest | 0.59 | 0.19 | 0.06 | 0.02 | 0.02 | 0.03 | 0.05 | 0.03 | 0.01 | 0.00 |
| Spc Invdrs | 0.42 | 0.18 | 0.11 | 0.06 | 0.04 | 0.02 | 0.02 | 0.02 | 0.03 | 0.10 |
| Star Gunner | 0.59 | 0.31 | 0.06 | 0.02 | 0.00 | 0.01 | 0.00 | 0.00 | 0.00 | 0.00 |
| Time Pilot | 0.580 | 0.14 | 0.11 | 0.05 | 0.03 | 0.01 | 0.01 | 0.01 | 0.02 | 0.04 |
| Tutankham | 0.16 | 0.74 | 0.02 | 0.01 | 0.01 | 0.01 | 0.01 | 0.01 | 0.01 | 0.01 |
| Wzd of Wor | 0.28 | 0.12 | 0.08 | 0.19 | 0.11 | 0.08 | 0.04 | 0.04 | 0.04 | 0.02 |

Table 8 contains the average action repetition chosen in each of the games for the two FiGAR-variants. The same episodes used to populate Table 6 and 7 were used to fill Table 8. It can be seen that in most games, the Stochastic variant of policy learns to play at a higher speed, although this might result in some loss in optimality of game play, as demonstrated in Figure 5.

Table 8: Average Action Repetition comparison between stochastic and greedy policies

| Name | Stochastic | 0.1-Greedy |
|---|---|---|
| Alien | 8.43 | 6.87 |
| Amidar | 13.77 | 9.61 |
| Assault | 7.14 | 5.86 |
| Asterix | 6.53 | 4.22 |
| Atlantis | 11.68 | 7.20 |
| Bank Heist | 1.65 | 1.62 |
| Beam Rider | 12.47 | 7.68 |
| Bowling | 28.64 | 5.13 |
| Breakout | 10.14 | 9.93 |
| Centipede | 6.84 | 7.88 |
| Chopper Command | 13.76 | 9.58 |
| Crazy Climber | 8.00 | 5.74 |
| Enduro | 2.91 | 2.69 |
| Demon Attack | 10.23 | 8.59 |
| Freeway | 14.62 | 14.25 |
| Frostbite | 11.33 | 7.69 |
| Gopher | 6.68 | 5.33 |
| James Bond | 14.98 | 10.37 |
| Kangaroo | 15.07 | 7.84 |
| Koolaid | 13.66 | 8.48 |
| Krull | 3.83 | 3.12 |
| Kung Fu Master | 10.00 | 8.53 |
| Name this Game | 14.98 | 9.55 |
| Phoenix | 10.31 | 4.64 |
| Pong | 12.99 | 12.28 |
| Q-bert | 2.02 | 1.76 |
| Road Runner | 1.63 | 1.26 |
| Sea Quest | 6.98 | 5.33 |
| Space Invaders | 10.48 | 8.55 |
| Star Gunner | 5.21 | 3.69 |
| Time Pilot | 12.72 | 5.39 |
| Tutankhamun | 9.75 | 5.73 |
| Wizard of Wor | 14.27 | 9.87 |

## APPENDIX C: EXPERIMENTAL SETUP FOR FiGAR-TRPO

### EXPERIMENTAL DETAILS

FiGAR-TRPO and the corresponding baseline algorithm operate on low dimensional feature vector observations. The TRPO (and hence FiGAR-TRPO) algorithm operates in two phases. In the first phase ($P1$), $K$ trajectories are sampled according to current behavioral policy $\pi$ to create the surrogate loss function. In the second phase ($P2$) a policy improvement step is performed by carrying out an optimization step on the surrogate loss function, subject to the KL-divergence constraint on the new policy. In our experiments, 500 such policy improvement steps were performed. $K$ varies with the learning progress and the schedule on what value $K$ would take in next iteration of $P1$ is defined linearly in terms of the return in the last iteration of $P1$. Hence if the return was large in previous iteration of $P1$, a small number of episodes are are used to construct the surrogate loss function in current iteration. The best policy was found by keeping track of the average returns seen during the training phase $P1$. This policy was then evaluated on 100 episodes to obtain the average score of the TRPO policy learnt. The most important hyper-parameters for FiGAR-TRPO are $\beta_{ar}$ and $\beta_{KL}$. By using a grid search on the set $\{0.01, 0.02, 0.04, 0.08, 0.16, 0.32, 0.64, 1.28\}$ we found the optimal hyper-parameters $\beta_{ar} = 1.28$ and $\beta_{KL} = 0.64$. These were tuned on all the 5 tasks.

LOSS FUNCTION AND ARCHITECTURE

The $\tanh$ non-linearity is used throughout. The mean vector is realized using a 2-Hidden Layer neural network (mean network) with hidden layer sizes $(128, 64)$. The standard deviation is realized using a Parameter layer (std-dev layer) which parameterizes the standard deviation but does not depend on the input. Hence the concatenation of the output of mean network and the std-dev layer forms the action policy $f_{\theta_a}$ as described in Section 4. The Action Repetition function $f_{\theta_x}$ is realized using a 2-Hidden Layer neural (act-rep network) network similar to the mean network albeit with smaller hidden layer sizes: $(128, 64)$. However, its output non-linearity is a softmax layer of size 30 as dictated by the value of $W$. The action repetition network was kept small to ensure that FiGAR-TRPO does not have significantly more parameters than TRPO. The mean network, std-dev layer and act-rep network do not share any parameters or layers (See appendix $G$ for experiments on FiGAR-TRPO with shared layers).

The surrogate loss function in TRPO when the Single Path method of construction is followed reduces to [Schulman et al. (2015)]:

$$L_{\theta_{old}}(\tilde{\theta}) = \mathbb{E}_{s \sim \rho^{\theta}_{old}, a \sim \pi_{\theta_{old}}} \left[ \frac{\pi_{\tilde{\theta}}(a|s)}{\pi_{\theta_{old}}(a|s)} Q_{\theta_{old}}(s, a) \right]$$

where $q$, the sampling distribution is just the old behavioral policy $\pi_{\theta_{old}}$ (defining characteristic of Single-Path method) and $\rho$ is the improper discounted state visitation distribution.

The surrogate loss function for a factored policy such as that of FiGAR-TRPO is:

$$L_{\theta_{a,old}, \theta_{x,old}}(\theta_a, \theta_x) = \mathbb{E}_{s,a,x} \left[ \frac{\pi_{\theta_a}(a|s)}{\pi_{\theta_{a,old}}(a|s)} \frac{\pi_{\theta_x}(x|s)}{\pi_{\theta_{x,old}}(x|s)} Q_{\theta_{a,old}, \theta_{x,old}}(s, a, x) \right]$$

where $s \sim \rho^{\theta_a, \theta_x}_{old}, a \sim \pi_{\theta_{a,old}}, x \sim \pi_{\theta_{x,old}}$ and $\pi_{\theta_a} = f_{\theta_a}$, $\pi_{\theta_{a,old}} = f_{\theta_{a,old}}$, $\pi_{\theta_x} = f_{\theta_x}$ and $\pi_{\theta_{x,old}} = f_{\theta_{x,old}}$

This kind of a splitting of probability distributions happens because the action-policy $f_{\theta_a}$ and the action-repetition policy $f_{\theta_x}$ are independent probability distributions. The theoretically sound way to realize FiGAR-TRPO is to minimize the loss $L_{\theta_{a,old}, \theta_{x,old}}(\theta_a, \theta_x)$. However, we found that in practice, optimizing a relaxed version of the objective function, that is,

$$L_{\theta_{a,old}, \theta_{x,old}}(\tilde{\theta}_a) \times L_{\theta_{a,old}, \theta_{x,old}}(\tilde{\theta}_x)^{\beta_{ar}}$$

works better. This leads to the FiGAR-TRPO objective defined in Section 4.3.

## APPENDIX D: EXPERIMENTAL DETAILS FOR FiGAR-DDPG

### EXPERIMENTAL DETAILS

The DDPG algorithm also operates on the low-dimensional (29 dimensional) feature-vector observations. The domain consists of 3 continuous actions, acceleration, break and steering. The $W$ hyper-parameter used in main experiments was chosen to be $15$ arbitrarily. Unlike Lillicrap et al. (2015), we did not find it useful to use batch normalization and hence it was not used. However, a replay memory was used of size 10000. Target networks were also used with soft updates being applied with $\tau = 0.001$. Sine DDPG is an off-policy actor-critic method, we need to ensure that sufficient exploration takes place. Use of an Ornstein-Uhlenbeck process (refer to Lillicrap et al. (2015) for details) ensured that exporation was carried out in action-policy space. To ensure exploration in the action-repetition policy space, we adopted two strategies. First, an $\epsilon$-greedy version of the policy was used during train time. The $\epsilon$ was annealed from $0.2$ to $0$ over $50000$ training steps. The algorithm was run for $40000$ training steps for baselines as well as FiGAR-DDPG. Second, with probability $1 - \epsilon$, instead of picking the greedy action-repetition , we sampled from the output distribution $f_{\theta_x}(s)$.

### ARCHITECTURAL DETAILS

Through the architecture, the hidden layer non-linearity used was ReLU. All hidden layer weights were initialized using the He initialization [He et al. (2015)]
The actor network consisted of a 2-hidden layer neural network with hidden sizes $(300, 600)$ (call the second hidden layer representation $h_2$. We learn two different output layers on top of this common hidden representation. $f_{\theta_a}$ was realized by transforming $h_2$ with an output layer of size 3. The output neuron corresponding to the action steering used $\tanh$ non linearity where as those corresponding to acceleration and break used the sigmoid non-linearity. The $f_{\theta_x}$ network was realized by transforming $h_2$ using a softmax output layer of size $|W|$. The output of the Actor network is a $3 + |W| = 18$ dimensional vector.
The critic network takes as input the state vector (29-dimensional) and the action vector (18-dimensional). The critic is a 3 hidden layer network of size $(300, 600, 600)$. Similar to Lillicrap et al. (2015), actions were not included until the $2^{nd}$ hidden layer of $f_{\theta_c}$. The final output is linear and is trained using the TD-error objective function, similar to Lillicrap et al. (2015)

APPENDIX E: DETAILS FOR FIGAR-VARIANTS

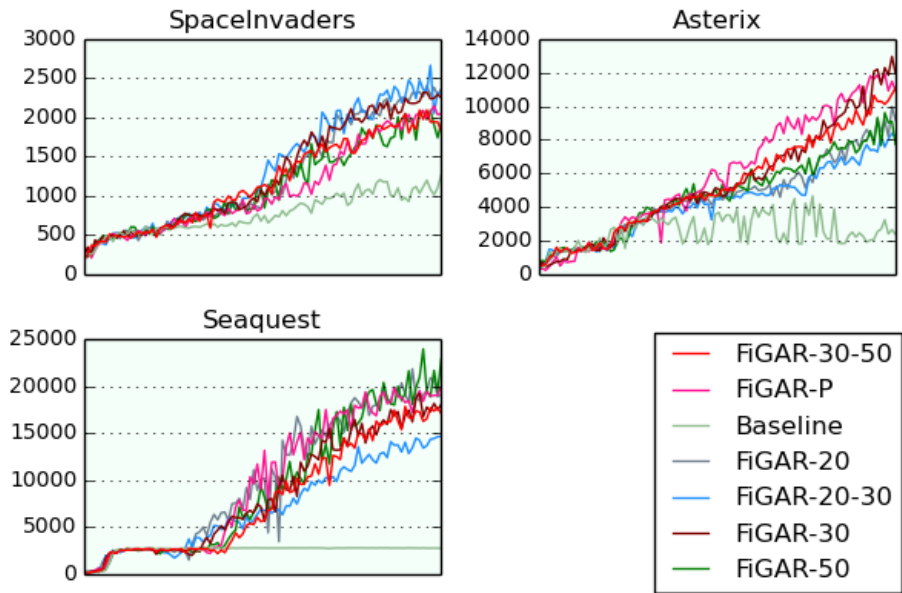

Figure 6: Comparison of FiGAR-A3C variants to the A3C baseline for 2 games: Sea Quest and Asterix

It is clear from Figure 6 that even though FiGAR A3C needs to explore in 2 separate action-spaces (those of primitive actions and the action repetitions), the training progress is not slowed down as a result of this exploration, for any FiGAR variant.

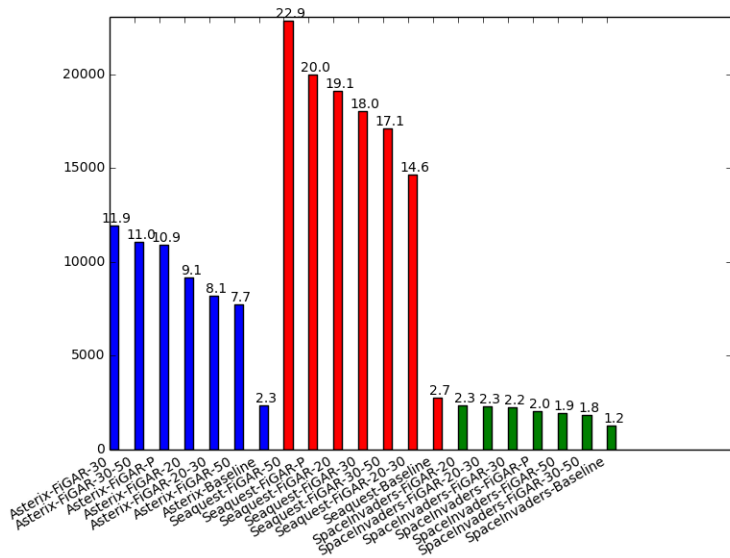

Figure 7: Comparison of FiGAR-A3C variants to the A3C baseline for 3 games: Sea Quest, Space Invaders and Asterix. Game scores have been scaled down by 1000 and rounded to 1 decimal place.

Table 2 contains final evaluation scores attained by various FiGAR variants. Figure 7 contains a bar-graph visualization of the same table to demonstrate the advantage of all FiGAR variants relative to the baselines.

APPENDIX F: IMPORTANCE OF $\pi_{\theta_x}$

One could potentially use FiGAR at evaluation stage (after training has been completed) at an action-repetition rate of 1 by picking every action according to $\pi_{\theta_a}$ and completely discarding the learnt repetition policy $\pi_{\theta_x}$. Such a FiGAR variant is denoted as FiGAR-wo-$\pi_{\theta_x}$. We demonstrate that FiGAR-wo-$\pi_{\theta_x}$ is worse than FiGAR on most games and hence the temporal abstractions learnt by and encoded in $\pi_{\theta_x}$ are indeed non-trivial and important for gameplay performance. Table 9 contains the comparison between standard FiGAR agent and FiGAR-wo-$\pi_{\theta_x}$. Evaluation scheme is the same as Appendix A.

Table 9: Gameplay performance of FiGAR compared with FiGAR-wo-$\pi_{\theta_x}$

| Name | FiGAR | FiGAR-wo-$\pi_{\theta_x}$ |
|---|---|---|
| Alien | **3138.50** | 582.17 |
| Amidar | **1465.70** | 497.90 |
| Assault | **1936.37** | 1551.40 |
| Asterix | **11949.00** | 780.00 |
| Atlantis | **6330600.00** | 680890.00 |
| Bank Heist | **3364.60** | 223.00 |
| Beam Rider | 2348.78 | **3732.00** |
| Bowling | **30.09** | 0.90 |
| Breakout | **814.50** | 321.90 |
| Centipede | 3340.35 | **3934.90** |
| Chopper Command | 3147.00 | 2730.00 |
| Crazy Climber | 154177.00 | 210.00 |
| Enduro | 707.80 | **941.10** |
| Demon Attack | 7499.30 | 6661.00 |
| Freeway | **33.14** | 30.60 |
| Frostbite | **309.60** | 308.00 |
| Gopher | **12845.40** | 10738.00 |
| James Bond | **478.0** | 320.00 |
| Kangaroo | **48.00** | 40.00 |
| Koolaid | 1669.00 | **2110.00** |
| Krull | 1316.10 | **2076.00** |
| Kung Fu Master | **40284.00** | 29770.00 |
| Name this Game | 1752.60 | 1692.00 |
| Phoenix | 5106.10 | **5266.00** |
| Pong | **20.32** | -21.00 |
| Road Runner | 22907.00 | **23560.00** |
| Sea Quest | 18076.90 | **18324.00** |
| Space Invaders | **2251.95** | 1721.00 |
| Star Gunner | 51269.00 | **55150.00** |
| Time Pilot | **11865.00** | 11810.00 |
| Tutankhamun | **276.95** | 182.20 |
| Wizard of Wor | **6688.00** | 6160.00 |

We observe that in 24 out of 33 games, $\pi_{\theta_x}$ helps the agent learn temporal abstractions which result in a significant boost in performance compared to the FiGAR-wo-$\pi_{\theta_x}$ agents.

APPENDIX G: SHARED REPRESENTATION EXPERIMENTS FOR FIGAR-TRPO

Section 5.2 contains results of experiments on FiGAR-TRPO. Appendix $C$ contains the experimental setup for the same. Throughout these experiments on FiGAR-TRPO the policy components $f_{\theta_a}$ and $f_{\theta_x}$ do not share any representations. This appendix contains experimental results in the setting wherein ($f_{\theta_a}$ and $f_{\theta_x}$) share all layers except the final one. This agent/network is denoted with the name FiGAR-shared-TRPO. All the hyper-parameters are the same as those in Appendix $C$ except $\beta_{ar}$ and $\beta_{KL}$ which were obtained through a grid-search similar to appendix $C$. These were tuned on all the 5 tasks. The values for these hyper-parameters that we found to be optimal are $\beta_{ar} = 1.28$ and $\beta_{KL} = 0.16$. The same training and evaluation regime as appendix $C$ was used. The performance of the best policy learnt is tabulated in Table 10

Table 10: Evaluation of FiGAR with shared representations for $f_{\theta_a}$ and $f_{\theta_x}$ on Mujoco

| Domain | FiGAR-TRPO | FiGAR-shared-TRPO | TRPO |
|---|---|---|---|
| Ant | 947.06 (28.35) | **1779.72** (7.99) | -161.93 (1.00) |
| Hopper | 3038.63 (1.00) | 2649.09 (2.07) | **3397.58** (1.00) |
| Inverted Pendulum | **1000.00** (1.00) | 986.35 (1.00) | 971.66 (1.00) |
| Inverted Double Pendulum | 8712.46 (1.01) | **9138.85** (1.00) | 8327.75 (1.00) |
| Swimmer | 337.48 (10.51) | 340.74 (8.02) | **364.55** (1.00) |

FiGAR-shared-TRPO on the whole does not perform much better than FiGAR-TRPO. In these TRPO experiments, the neural networks we used were rather shallow at only two hidden layers deep. Hence, we believe that sharing of layers thus leads to only small gains in terms of optimality of policy learnt.

