# Peer review of "Learning to Repeat: Fine Grained Action Repetition for Deep Reinforcement Learning"

_ICLR 2017 — accepted_

[Public Comment · Aravind Rajeswaran · 15 Dec 2016 (modified: 16 Dec 2016)]
**some questions and comments**

Hi, the main idea is quite interesting. I was curious about the following. My primary question is Q1, and others are predominantly comments.

Q1: After learning is complete, did you try forward propagating through the network to find actions for every time-step as opposed to repeating actions? Concretely, if at t=5, action suggested by the network is a_3 with a repetition of 4, instead of sticking with a_3 for times t={5,6,7,8} perform action a_3 for just t=5, and forward prop through the policy again at t=6.

I understand that the goal is to explore temporal abstractions, but for all the problems considered in this paper, a forward prop is not expensive at all. Hence, there is no computational bottle neck forcing action repetition during test-time. It is understandable that repeating actions speeds up training. However, at test time, the performance can potentially improve by not repeating. This idea is quite popular in variants of Receding Horizon Control and MCTS.

2: Can you share hyper-parameter settings of section 5.2? How many iterations of TRPO was run, and how many trajectory samples per iteration? The performance on Ant-v1 task is too low for both TRPO and FIGAR. Running for more iterations and initializing the network better (smaller weights) might improve performance significantly for both. It might be informative to share the learning curves comparing FIGAR with TRPO. With the current results, it is a stretch to say that FIGAR "outperforms" TRPO.

3: Considering that the action repetition is 1 for a majority of MuJoCo tasks (discounting Ant) and TORCS, why do you expect FIGAR to perform better? Also, the FIGAR policies seem to have more parameters than baselines they are compared against -- is this true? Have you compared to baselines with equal number of parameters?

[Official Review · AnonReviewer2 · rating 7 · confidence 3 · 17 Dec 2016]

This paper provides a simple method to handle action repetitions. They make the action a tuple (a,x), where a is the action chosen, and x the number of repetitions. Overall they report some improvements over A3C/DDPG, dramatic in some games, moderate in other. The idea seems natural and there is a wealth of experiment to support it.

Comments:

- The scores reported on A3C in this paper and in the Mnih et al. publication (table S3) differ significantly. Where does this discrepancy come from? If it's from a different training regime (fewer iterations, for instance), did the authors confirm that running  their replication to the same settings as Mnih et al provide similar results?

- It is intriguing that the best results of FiGAR are reported on games where few actions repeat dominate. This seems to imply that for those, the performance overhead of FiGAR over A3C is high since A3C uses an action repeat of 4 (and therefore has 4 times fewer gradient updates). A3C could be run for a comparable computation cost with a lower action repeat, which would probably result in increased performance of A3C.  Nevertheless,  the automatic determination of the appropriate action repeat is interesting, even if the overall message seems to be to not repeat actions too often.

- Slightly problematic notation, where r sometimes denotes rewards, sometimes denotes elements of the repetition set R (top of page 5)

- In the equation at the bottom of page 5 - since the sum is not indexed over decision steps, not time steps, shouldn't the rewards r_k be modified to be the sum of rewards (appropriately discounted) between those time steps?

- The section on DDPG is confusingly written. "Concatenating" loss is a strange operation; doesn't FiGAR correspond to a loss to roughly looks like Q(x,mu(x)) + R log p(x) (with separate loss for learning the critic)? It feels that REINFORCE should be applied for the repetition variable x (second term of the sum) and reparametrization for the action a (first term)? 

- Is the 'name_this_game' name in the tables  intentional?

- A potential weakness of the method is that the agent must decide to commit to an action for a fixed number of steps, independently of what happens next. Have the authors considered a scheme in which, at each time step, the agent decides to stick with the current decision or not? (It feels like it might be a relatively simple modification of FiGAR).

[Official Review · AnonReviewer4 · rating 8 · confidence 4 · 17 Dec 2016]

This paper proposes a simple but effective extension to reinforcement learning algorithms, by adding a temporal repetition component as part of the action space, enabling the policy to select how long to repeat the chosen action for. The extension applies to all reinforcement learning algorithms, including both discrete and continuous domains, as it is primarily changing the action parametrization. The paper is well-written, and the experiments extensively evaluate the approach with 3 different RL algorithms in 3 different domains (Atari, MuJoCo, and TORCS).

Here are some comments and questions, for improving the paper:

The introduction states that "all DRL algorithms repeatedly execute a chosen action for a fixed number of time steps k". This statement is too strong, and is actually disproved in the experiments — repeating an action is helpful in many tasks, but not in all tasks. The sentence should be rephrased to be more precise.

In the related work, a discussion of the relation to semi-MDPs would be useful to help the reader better understand the approach and how it compares and differs (e.g. the response from the pre-review questions)

Experiments:
Can you provide error bars on the experimental results? (from running multiple random seeds)

It would be useful to see experiments with parameter sharing in the TRPO experiments, to be more consistent with the other domains, especially since it seems that the improvement in the TRPO experiments is smaller than that of the other two domains. Right now, it is hard to tell if the smaller improvement is because of the nature of the task, because of the lack of parameter sharing, or something else.

The TRPO evaluation is different from the results reported in Duan et al. ICML ’16. Why not use the same benchmark?

Videos only show the policies learned with FiGAR, which are uninformative without also seeing the policies learned without FiGAR. Can you also include videos of the policies learned without FiGAR, as a comparison point?

How many laps does DDPG complete without FiGAR? The difference in reward achieved seems quite substantial (557K vs. 59K).

Can the tables be visualized as histograms? This seems like it would more effectively and efficiently communicate the results.

Minor comments:
-- On the plot in Figure 2, the label for the first bar should be changed from 1000 to 3500.
-- “idea of deciding when necessary” - seems like it would be better to say “idea of only deciding when necessary"
-- "spaces.Durugkar et al.” — missing a space.
-- “R={4}” — why 4? Could you use a letter to indicate a constant instead? (or a different notation)

[Official Review · AnonReviewer3 · rating 8 · confidence 5 · 18 Dec 2016 (modified: 20 Jan 2017)]
**Simple but effective idea with a very thorough evaluation**

This paper shows that extending deep RL algorithms to decide which action to take as well as how many times to repeat it leads to improved performance on a number of domains. The evaluation is very thorough and shows that this simple idea works well in both discrete and continuous actions spaces.

A few comments/questions:
- Table 1 could be easier to interpret as a figure of histograms.
- Figure 3 could be easier to interpret as a table.
- How was the subset of Atari games selected?
- The Atari evaluation does show convincing improvements over A3C on games requiring extended exploration (e.g. Freeway and Seaquest), but it would be nice to see a full evaluation on 57 games. This has become quite standard and would make it possible to compare overall performance using mean and median scores.
- It would also be nice to see a more direct comparison to the STRAW model of Vezhnevets et al., which aims to solve some of the same problems as FiGAR.
- FiGAR currently discards frames between action decisions. There might be a tradeoff between repeating an action more times and throwing away more information. Have you thought about separating these effects? You could train a model that does process intermediate frames. Just a thought.

Overall, this is a nice simple addition to deep RL algorithms that many people will probably start using.

--------------------

I'm increasing my score to 8 based on the rebuttal and the revised paper.

[Author Response · Sahil Sharma · 16 Jan 2017]
**Revision in response to reviewer comments and questions**

We thank all the reviewers for asking interesting questions and pointing out important flaws in the paper. We have uploaded a revised version of the paper that we believe addresses the questions raised. Major features of the revision are:

1. We have added results on 2 more Atari 2600 games: Enduro and Q-bert. FiGAR seems to improve performance rather dramatically on Enduro with the FiGAR agent being close to 100 times better than the baseline A3C agent. (Note that the baseline agent performs very poorly according to the published results as well)

2. In response to AnonReviewer3’s comment about skipping intermediate frames, we have added Appendix F (page 23) by conducting experiments on what happens when FiGAR does not discard any intermediate frames (during evaluation phase). The general pattern seems to be that for games wherein lower action repetition is preferred, gains are made in terms of improved gameplay performance. However, for 24 out of 33 games the performance becomes worse, which depicts the importance of the temporal abstractions learnt by the action repetition part of the policy (\pi_{\theta_{x}}). This does not address the reviewer’s question completely since at train time we still skip all the frames, as suggested by the action repetition policy. We have added a small discussion on future works section (section 6, page 10) which could potentially address this comment.

3. In response to AnonReviewer3’s suggestion to turn table1 into a bar graph we have done so (Figure 3, page 8) and it indeed does look much better.

4. In response to AnonReviewer3’s suggestion to compare directly to STRAW we have added Table 5 (Appendix A, page 14) which contains performance of STRAW models on all games which we have also experimented with. The general conclusion seems to be that in some games STRAW does better and in some games FiGAR does better.

5. In response to AnonReviewer4’s comment, we conducted experiments on shared representations for the FiGAR-TRPO agent. Appendix G (page 24) contains the results of the experiments. In general we observe that FiGAR-TRPO with shared representations does marginally better than FiGAR-TRPO, but not much better. The performance goes down on some tasks and improves on others. The average action repetition rate of the best policies learnt improves.

6. In response to AnonReviewer4’s comment on SMDPs we have added the relevant discussion to related works section (page 3).

7. In response to AnonReviewer2’s comment on the confusing nature of FiGAR-DDPG section, we have rewritten the section. It is hopefully clearer now.
 
8. In response to AnonReviewer2’s comment on the confusing notation ‘r’ for action repetition we have completely changed the notation for action repetition to the letter ‘w’.

9. In response to AnonReviewer2’s comment on  the potential weakness of the FiGAR framework, we have added a discussion on the shortcomings of the FiGAR in section 6 (page 10).

10.We have corrected several typos as pointed out by the reviewers.

[Final Decision · Program Chairs · 06 Feb 2017]
**ICLR committee final decision**

The basic idea of this paper is simple: run RL over an action space that models both the actions and the number of times they are repeated. It's a simple idea, but seems to work really well on a pretty substantial variety of domains, and it can be easily adapted to many different settings. In several settings, the improvement using this approach are dramatic. I think this is an obvious accept: a simple addition to existing RL algorithms that can often perform much better.
 
 Pros:
 + Simple and intuitive approach, easy to implement
 + Extensive evaluation, showing very good performance
 
 Cons:
 - Sometimes unclear _why_ certain domains benefit so much from this